# Traffic Noise and Ambient Air Pollution Are Risk Factorsfor Preeclampsia

**DOI:** 10.3390/jcm11154552

**Published:** 2022-08-04

**Authors:** Natalya Bilenko, Michal Ashin, Michael Friger, Laura Fischer, Ruslan Sergienko, Eyal Sheiner

**Affiliations:** 1Department of Epidemiology, Biostatistics and Community Health Sciences, School of Public Health, Faculty of Health Sciences, Ben-Gurion University of the Negev, Beer Sheva 8410501, Israel; 2Ministry of Health, Medical Office of Ashkelon District, Ashkelon 78306, Israel; 3Department of Obstetrics and Gynecology, Soroka University Medical Center, School of Public Health, Faculty of Health Sciences, Ben-Gurion University of the Negev, P.O. Box 151, Beer Sheva 8410501, Israel

**Keywords:** traffic noise, ambient air pollution, NOx, preeclampsia, prenatal exposure

## Abstract

Purpose: We aimed to evaluate the effect of traffic-related noise (TRN), environmental noise (EN) and traffic-related air pollution (TRAP) on preeclampsia. ***Methods:*** We followed 285 pregnant women from Maternal and Child Health Clinics who reported exposure to TRN on a scale from 0 (absence of EN) to 10 (high level of EN). EN was measured using a portable dosimeter, and NOx was calculated using the AERMOD pollutant dispersion model. ***Results:*** Using a multiple logistic regression model, adjusted for maternal age, BMI, number of births, fetal sex and maternal chronic illness, TRN (score ≥ 6 vs. score < 6) and TRAP (NOx ≥ 300 µ/m^3^ vs. NOx < 300 µ/m^3^) were noted as independent risk factors for preeclampsia, with OR = 3.07 (95% CI 0.97; 9.70, *p* = 0.056) and OR = 3.43 (95% CI 1.20; 9.87, *p* = 0.022), respectively. Joint exposure to TRN and TRAP was associated with a significant and independent risk for preeclampsia (OR of 4.11 (95% CI 1.31; 12.94, *p* = 0.016). ***Conclusions:*** In our population, traffic-related noise and ambient TRAP were risk factors for preeclampsia.

## 1. Introduction

A wide variety of sources of environmental hazards exist in the modern world, originating from traffic (air, road or rail) and industrial facilities [1,2]. Road traffic is the main source of both air and noise pollution with a negative effect on human health. Noise exposure is associated with several adverse health effects [2,3,4,5,6,7,8]. The impact of noise on human health depends on several factors, including the level of noise, the person who is exposed and the exposure duration [1,2]. Noise can reduce sleep quality as well as cause physiological, mental and social effects, including: impatience, aggression and difficulty in making decisions [3,4]). Environmental noise exposure has also been associated with progression of cardiovascular disease and can result in an increase in hypertension [5,6,7,8]. For example, the OR for hypertension was 1.9 (95% CI 1.1 to 3.5) in the highest noise category (56–70 dB(A)) and 3.8 (95% CI 1.6 to 9.0) in men [6]. The authors concluded that traffic noise heard in urban environments is a significant factor contributing to sleep disorders, heart and blood vessel diseases [6].

Several studies have demonstrated that community noise may also have adverse effects on reproductive outcomes. One cross-sectional study on pregnant women in Lithuania documented a relationship between a high level of traffic noise (above 61 dB (A)), and gestational hypertension (OR 1.36, 95% CI 0.86; −2.15) [9]. An additional harmful effect of road traffic is air pollution, which contributes to the increased risk of hypertension and other cardiovascular events in adults [10,11,12].

A limited number of studies have examined the association between traffic noise, ambient air pollution and preeclampsia [13,14]. Thus, the current study aimed to examine whether traffic-related noise and air pollution are associated with increased incidences of preeclampsia.

## 2. Methods

### 2.1. Study Population

In this prospective cohort study, 316 pregnant women who underwent prenatal follow up at the Maternal and Child Health Clinics (MCHC) of Ministry of Health in Beer Sheva were recruited. In total, 285 (90%) of the women who were approached agreed to participate in the study. Women with multiple pregnancies and those who were not planning to give birth at the Soroka University Medical Center, the only tertiary hospital in the Negev, as well as those diagnosed with chronic hypertension, were excluded from the study. Data on birth outcomes were available for 253 women (88.8%). Beer Sheva is the capital of southern Israel, a city with a mostly Jewish population at an intermediate SES level. We recruited women weekly on randomly chosen days in eight Beer Sheva MCHCs, from women who arrived to MCHCs to register for prenatal follow up. This was a representative sample of Israeli Jewish women, since there is no difference between rural and urban Jewish settlements in Israel in terms of both life style and preventive and curative health care utilization. The study was performed between 2013 and 2016, while 93% of participants were recruited from the end of 2014 to the end of 2015. We did assume that there were no fluctuations in study years in exposure levels regarding air pollution and noise, since no road constructions changed and no new manufacturing buildings were raised.

### 2.2. Exposure Assessment

In this study, we examined two types of traffic-related environmental exposure: noise and air pollution. 

#### 2.2.1. Noise Exposure

Noise exposure assessment was performed in two ways: (1) self-reported traffic-related noise (TRN) annoyance and subjective effect on sleep quality, and (2) measured environmental noise (EN).

***Self-reported TRN annoyance and its subjective effect on sleep quality*** were assessed through personal interviews between the years 2013 and 2016. Participants were asked about their subjective sensitivity to noise in their living environment during pregnancy at enrollment during routine prenatal visits in MCHCs. The women reported the level of TRN annoyance and the impact on quality of sleep in their homes on a scale (between 0—no noise at all and 10—the highest level of noise) using a validated tool [15]. Women with a TRN score ≥ 6 were defined as being affected by TRN. In addition, this questionnaire included personal information on socio-demographic information, reported BMI before pregnancy, obstetric history, chronic illness (cardiovascular and /or respiratory disease and mental stress before pregnancy), BMI before pregnancy, smoking status, alcohol consumption and description of living environment.

***Measured EN*** was assessed using a portable dosimeter at the facade at the home address of the participants. Noise level measurements were evaluated using the SLM SPARK 706 noise meter. Noise level was measured in decibels (dB(A)). Minimum, maximum and average noise values were taken at each residential address during pregnancy. The environmental noise was measured between February 2016 and June 2016 between 16:00 and 19:00. Each measurement lasted 10 min. During these hours, the noise level is higher and relatively constant, and 10 min measurements of the noise instrument indicate a relatively representative noise level of the measured area [16]. Average noise level at each residence was recorded, as well as minimum and maximum levels. Increased traffic load hours were noted to be from 7:00 am to 10:00 a.m., and again from 16:00 to 19:00.

Self-reported noise exposure was assessed during the first interview, assuming that there were no changes in the level of noise during the study period. Environmental noise exposure was measured in the evening hours, which were routinely the hours that our participants returned home from work. None of our participants changed residence during the study period.

#### 2.2.2. Traffic-Related Air Pollution (TRAP) Level

TRAP level was calculated using the “AERMOD pollutant dispersion model” [17]. The maximum concentrations of NOx in the area were calculated using a pollutant dispersal model according to Israeli Ministry of Environmental Protection. The main source of nitrogen oxide (NOx) in the air is related to transportation [18]. The AERMOD model can be used to simulate convection and dispersion of pollutants from several sources. This distribution is calculated, among other things, based on a recent characterization of the planetary boundary layer. The location of the emission sources can be urban or rural, and receptors can be placed in flat or complex topography. Using the PRIME algorithm, the AERMOD model can take into the account the effect of buildings and their proximity to emission sources. The model uses metrological data and works in conjunction with two data processors: AERMET, which processes meteorological data, and AERMAP, which processes topographic data [17].

The calculation was performed based on meteorological data that were measured in the Beer Sheva monitoring station of the Ministry of Environmental Protection during the years 2010–2014. The sources of emissions that were included in the calculation were transportation and central emission sources in the Beer Sheva area. Emission data were obtained from the Ministry of Environmental Protection.

The method of calculating transport sources was performed using a nitrogen oxide (NOx) emissions file received from the Ministry of Environmental Protection, which specified the maximum hourly emissions from road segments in the Beer Sheva area. The emission values were corrected for a 24 h time frame. Women with NOx levels ≥ 300 µ/m^3^ were defined as having been exposed to air pollution.

Figure 1 shows a map presenting the maximum concentrations calculated in Beer Sheva using lines of equal value (isopleths). This type of map shows gradual change over space and avoids abrupt changes. Using the ArcGIS software, the addresses of the participants were inserted, which made it possible to estimate the concentrations of NOx to which each woman was exposed.

***Joint exposure***—the composite variable describes exposure to both reported transport noise (6+) and an average level of measured NOx air pollution above 300 µ/m^3^ (n = 33).

### 2.3. Outcome

Variables were defined using diagnoses from hospitalization files coded using ICD-9. Those with codes 642.4 and 642.5 es were defined as having “mild preeclampsia” and “severe preeclampsia”, respectively.

### 2.4. Statistical Analysis

Univariate associations between exposure to noise and air pollution and mild and severe preeclampsia were examined using a Pearson correlation test, a *t*-test and a Mann–Whitney rank test for variables without distribution (number of births). Due to the low number of cases of severe and mild preeclampsia, we combined them into one group named preeclampsia. Traffic noise (dB(A)) and NOx concentration (µg/m^3^) were entered into the model as continuous variables. Age, pre-pregnancy BMI, parity and chronic morbidity of mother were entered into the multivariate logistic regression model analysis as potential confounders. Interaction analysis was carried out to explore the individual and joint effects of the two exposures. *p* of <0.05 was considered statistically significant. Statistical analysis of data was performed using the SPSS software program (version 24; SPSS Inc., Chicago, IL, USA).

## 3. Results

This study followed 285 pregnant women from the moment of registration for routine prenatal care in MCH clinics until the delivery. At the end of follow up, medical records were available only for 253 (88.8%) women—the final study population. The average age of the study women was 31.2 ± 4.9 years, with a reported average pre-pregnancy Body Mass Index (BMI) of 23.9 ± 4.8. About one-quarter of the participants reported that they lived near a highway, and 53% of the women reported they lived near a bus station. 

Twenty-four women (9.5%) were diagnosed with preeclampsia (7.1% mild and 2.4% severe). Table 1 presents background socio-demographic and health characteristics of the study population with preeclampsia (yes/no). Women with preeclampsia had a higher pre-pregnancy BMI (*p* = 0.057), about 5 times the percentage of those with chronic diseases and a lower number of births (*p* = 0.007).

The average level of environmental noise measured by dosimeter at the women’s home addresses was 64 ± 6.9 dB (A). The average level of environmental noise was measured at 238 (94.1%) addresses out of the final study population.

The average score of self-reported traffic-related noise was 5.1 ± 2.9 with a median of 5. The correlation between the reported noise and the measured noise in the total study population was significant but relatively low (r = 0.21, *p* = 0.001). When we divided the women into two strata according to their reported living environment, the correlation coefficients were: 0.25 (*p* < 0.001) for women living near main roads, 0.30 (*p* < 0.001) for those living near schools and 0.43 (*p* < 0.001) within strata of women living near bus stops.

The average of the half-hour concentrations of nitrogen oxides at the 224 addresses of the women was 245.9 ± 83.2 (μg/m^3^). Figure 2 and Figure 3 show box-plot diagrams of the distribution of NOx concentrations at the participants’ residential addresses.

We further tested the association between preeclampsia and measured and reported traffic-related noise. We did not find a statistically significant association between measured noise level and preeclampsia in the participants. However, reported traffic noise was significantly higher in women who subsequently developed preeclampsia (6.4 ± 2.4 vs. 4.8 ± 2.9, *p* = 0.014). The noise reported as 6 or higher was a risk factor for the development of preeclampsia (unadjusted OR = 1.13; 95% CI 1.1–1.2, *p* = 0.004).

Using a multiple logistic regression, we examined the individual and joint effects of noise and air pollution on preeclampsia. In this analysis, we had data on both exposures for 217 participants, of which 63 women were classified as having no exposure, 28 only air pollution exposure, 93 only noise exposure and 33 both air pollution and noise exposure.

The adjusted OR was 3.07 (96% CI 0.97; 9.70, *p* = 0.056) for the association between ***only noise exposure*** and preeclampsia and 3.44 (95% CI 1.20; 9.87) between ***only air pollution*** and preeclampsia (Table 2). Exposure to both noise and air pollution (as compared to women with no exposure) was associated with a 4-fold increase in risk, with an OR of 4.11 (95% CI 1.31; 12.94, *p* = 0.016). The interaction term was not statistically significant.

## 4. Discussion

Exposure to traffic noise and ambient air pollution in the residential environment has become an integral part of our lives. The density of the population and the concentration of traffic volume have increased in western society over the years. The impact of TRN and TRAP on human health, especially morbidity of heart disease, stroke and hypertension, is remarkable. [5,6,7,8,9,10,11,12]. Our prospective study focused on examining the relationship between exposure to both traffic self-assessed noise and government-monitored air pollution and the health of pregnant women (mild or severe preeclampsia). Very few studies have examined the effect of noise and air pollution on this vulnerable population [19]. Auger et al. reported that the prevalence of preeclampsia was higher for women exposed to elevated environmental noise pollution levels (LAeq24h ≥ 65 dB(A) = 37.9 per 1000 vs. <50 dB(A) = 27.9 per 1000). Associations were, however, present with severe (OR 1.29, 95% CI 1.09–1.54) and early-onset (OR 1.71, 95% CI 1.20–2.43) preeclampsia, with consistent results across all noise indicators. 

The major finding of our study was an association between TRN and TRAP and preeclampsia in the study population. The rate of preeclampsia in our study population was 9.5%, in accordance with other studies [20,21,22]. Previous studies have suggested an association between preeclampsia and environmental factors such as seasonal patterns [23] and environmental pollutions [24].

This study focused on traffic noise and air pollution in the residential environment.

The correlation between the reported noise and the measured noise was significant, but relatively low in our study. This is probably because the women were asked about their sensitivity to traffic noise, and the measured noise was from general environmental noise. Obviously, it was difficult to separate between noise types within the residential environment. The women were asked in the questionnaire about additional sources of noise, including exposure to noise at work and shift work, which could be an additional stressor [25]. They were also asked guiding questions to accurately describe the area in which they lived, to locate the sources of noise to which they were exposed. According to the questionnaire, the correlation between the general measured noise level and the reported sensitivity to traffic noise was increased. Other reasons for the low correlation are that the sensitivity to noise varies from woman to woman and is not always dependent on the actual noise level. What primarily affects a woman’s health may be the feeling that the traffic noise causes, as disturbing noise can induce stress. Since the noise was not measured indoors but outside the home, it was difficult to accurately measure the level of noise that each woman experienced. Indeed, other studies using both methods of noise level estimation had difficulties isolating the traffic noise during measurement, and the correlation between measured noise at night and reported noise was relatively low (r = 0.135) [26].

In this study, a model was used for the assessment of air pollution levels due to traffic. There have been studies examining air pollution from sandstorms in the Negev region that affect air pollution in Beer Sheva [27]. However, traffic volumes in Beer Sheva are not large, and accordingly, nitrous oxide levels measured in the two monitoring stations installed in the city do not pass the established standards. The highest concentration calculated by the model was 415 µ/m^3^, which is less than 50% of the permitted standard.

The weak but significant correlation between NOx and general environmental noise level that was measured using a mobile device can be attributed to the differences between measurement methods (model vs. noise instrument) as well as the lack of coordination between measurement times. Several studies found a higher correlation by using the same measurement method in the same place, on the same day [16,28]. In another study, both noise and air pollution levels were assessed using questionnaires, and the correlation was high (0.61) [29].

In a study conducted in Lithuania [9], an association was found between traffic noise and gestational hypertension, but the association with air pollution from transportation was not tested for. A study in the United States found an association between the development of preeclampsia following exposure to nitrogen dioxide and fine particles (PM2.5), but the issue of noise was not examined [30]. A study conducted in 2017 in Denmark examined the effects of environmental air pollution and traffic noise on preeclampsia and hypertensive disorders. The authors of this article found a positive association between exposure to NO_2_ during the first trimester of pregnancy and risk of preeclampsia [31].

It appears that the population of pregnant women is more sensitive than the general population to the effects of both noise and air pollution. Noise that was documented as a risk factor for cardiovascular disease (including hypertension and ischemic heart disease) is also a stressful psychological factor affecting both the sympathetic nervous system and the endocrine system. Certain human actions, such as concentration, relaxation or sleep, are disturbed by noise; thus, exposure to traffic noise can release stress hormones that may increase blood pressure [32,33]. Pressure hormones such as adrenaline, norepinephrine and cortisone are used as reliable stress indicators [34,35]. They are part of positive and negative feedback mechanisms that affect heart activity, blood pressure, blood lipids and blood sugar, which are biological risk factors for conditions such as hypertension, atherosclerosis and myocardial infarction [36].

Experimental studies with animal models found a significant increase in the number of abnormal embryos in mice exposed to a noisy environment. Another study found a significant association between the reproductive efficiency of mice and noise level within one hour of exposure to 70 dB [37]. Various studies on the impact of air pollution on women’s health during pregnancy have investigated the mechanisms and biological actions that affect the health of women and fetuses during pregnancy. Several studies have shown the detrimental effects of environmental pollution on the developing fetus, which is more sensitive than the mother, and have further demonstrated that exposure to environmental infections may be biologically significant even in low doses [38]. The possible mechanism for the effect of carbon monoxide (CO) on embryos is related to its ability to bind to hemoglobin in embryos at a 10-times-higher rate than in adults, and therefore its evacuation ability is slower. When CO crosses the placenta, it damages the oxygen transfer capacity in the embryo, and intrauterine growth may be impaired, as is the case with fetal exposure to cigarette smoke. In addition, exposure to particles may also affect the effectiveness of placental function and damage the blood flow of the cord. Very small particles can cause the development of inflammatory reactions and thus reduce blood flow to the placenta. Placental dysfunction may be a major factor in the development of preeclampsia [39].

Our study has several strengths that stem from its prospective nature and the thorough questionnaires that were administered. A personal interview creates a better interaction with the interviewee and provides deeper answers. However, it has several weaknesses, including possible recall bias, which could underestimate its effect, and the fact that the noise tests were limited only to the women’s residential addresses. Noise levels were not measured at locations they may have often occupied outside of the home, such as for work and leisure. However, using validated questionnaires for self-reported noise may overcome this problem.

In addition, exposure to air traffic pollution was based solely on a model built for concentrations of nitrogen oxides. Data on benzene, carbon monoxide or ozone were not available; therefore, general nitrogen oxides and nitrogen dioxide could not be separated from the model. Although the model is applied only to nitrogen oxides (NOxs), it is still the best measure of transport air pollution, providing a reliable estimate of the levels of air pollution due to transportation in Beer Sheva. There is a problem with the separation of particles from other sources in the Negev and particulate matter from air pollution due to transportation. There is also difficulty in separating the noise measurements for noise from transportation and general environmental noise.

We found a 3–4-fold increase in the risk for preeclampsia among women exposed to individual noise and air pollution, as well as to both exposures without interactions. The small number exposed to both air pollution and noise exposure (which were part of the limitations of this study) resulted in a wide 95% confidence interval.

In conclusion, traffic-related noise and ambient air pollution are possible risk factors for the development of preeclampsia. Further large-scale studies should investigate these associations with additional measurements of stress mediators. In addition, such studies might measure noise and air pollution at the same time of the year, preferably on the same day, to neutralize any meteorological effects.

## Figures and Tables

**Figure 1 jcm-11-04552-f001:**
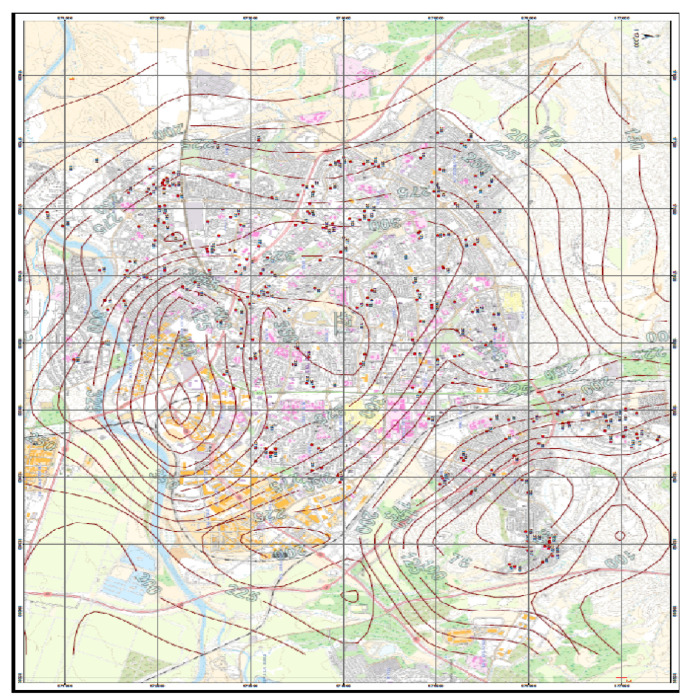
Half-hour average concentration of nitrogen oxides in Beer Sheva calculated by AIRMOD pollutant dispersion model with the addresses of the women (red dots).

**Figure 2 jcm-11-04552-f002:**
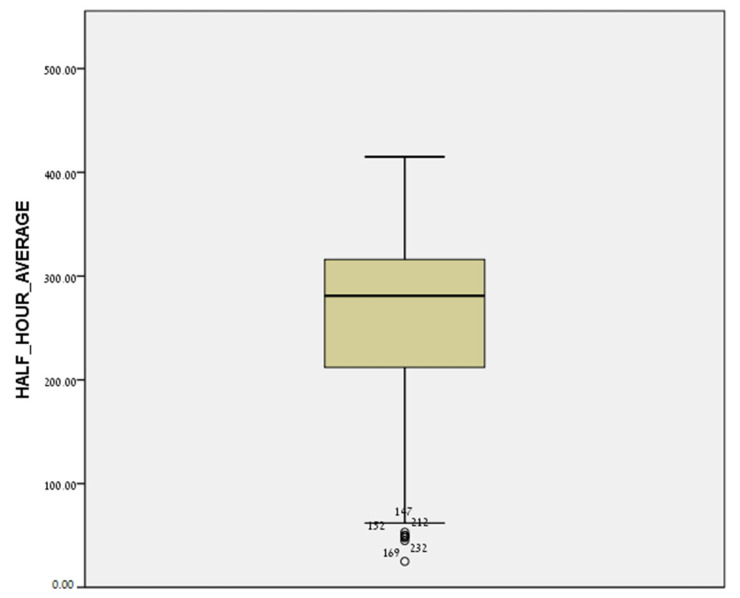
Distribution of NOx concentrations at the participants’ residential addresses (Box-plot diagram).

**Figure 3 jcm-11-04552-f003:**
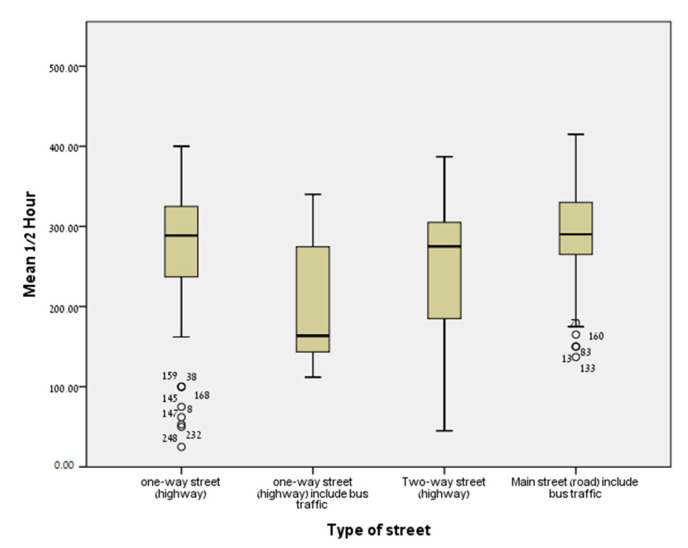
Distribution of NOx concentrations at the participants’ residential addresses by type of street (box-plot diagrams).

**Table 1 jcm-11-04552-t001:** Background socio-demographic and health characteristics of the study population.

Characteristics	Preeclampsia		*p*
	Yes (n = 24)	No (n = 229)	
	n (%) or Mean ± SD		
Maternal age, y (mean ± SD)	31.3 ± 5.7	31.3 ± 5.0	0.992 ^1^
BMI before pregnancy	25.8 ± 5.7	23.8 ± 4.7	0.057 ^1^
Number of birth	1.8 ± 0.8	2.8 ± 1.8	0.007 ^2^
Education (years)	14.5 ± 2.5	14.7 ± 2.7	0.525 ^1^
Pregnancy week	33.5 ± 3.7	31.0 ± 3.1	0.249 ^1^
Fetal sex			0.159 ^3^
Male	15 (10.8)	124 (89.2)	
Female	9 (6.2)	137 (93.8)	
Chronic Illness *, yes	6 (33.3)	17 (7.4)	<0.001 ^3^

* Chronic Illness: cardiovascular and/or respiratory disease, *p* values of ^1^
*t*-test, ^2^ Mann–Whitney test, ^3^ chi-square test.

**Table 2 jcm-11-04552-t002:** Multivariate logistic regression for the prediction of preeclampsia by joint air pollution and noise exposure.

Variable	OR	95% CI	*p*
Joint air pollution and noise exposure (Exposed to both noise and air pollution vs. non-exposed)	4.11	1.31; 12.94	0.016
Age, years	1.06	0.95; 1.16	0.363
BMI before pregnancy	1.08	0.97; 1.19	0.169
Number of births	0.48	0.28; 0.83	0.008
Fetal sex (male vs. female)	1.98	0.69; 5.70	0.208
Chronic illness (yes vs. no)	8.27	1.93; 23.01	0.004

## Data Availability

Not applicable.

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
