# Peer review of "Traffic Noise and Ambient Air Pollution Are Risk Factorsfor Preeclampsia"

_jcm, 2022, doi:10.3390/jcm11154552_

Round 1
Reviewer 1 Report
The study was aimed to evaluate effect of traffic-related noise (TRN), environmental noise (EN) and 20 traffic related air pollution (TRAP) on the outcome of preeclampsia. The authors followed 285 pregnant women from Maternal and Child Health Clinics and reported exposure to TRN on a scale from 0 (absence of EN) to10 (high level of EN). EN was measured using a portable dosimeter, and NOx was calculated using the AERMOD pollutant dispersion model. Exposure to high self-reported TRN (score>6) was found as a risk factor for preeclampsia (OR=1.13, p=0.004). They adjusted the results using multiple logistic regression model for maternal age, BMI, number of births, fetal sex and maternal chronic illness. They reported that TRN and elevated NO were risk factor for preeclampsia. Moreover, they reported that combined exposure to TRN and TRAP compounded the increased risk for preeclampsia. The authors concluded that Traffic-related noise and ambient TRAP are independent risk factors for preeclampsia.
The study is interesting and well written and is important in its novelty which surely would be cited extensively. I applaud the authors for such an impressive work and recommend acceptance following few minor changes:
abstract: please be more modest with the conclusion toning down. Indeed Traffic-related noise and ambient TRAP were risk factors for preeclampsia. I would add something like: In our population, traffic related noise and ambient TRAP were risk factors for preeclampsia.
Table 1 the sex distribution is unclear. What does the percentage represent?
Likewise, I would present the females as well
Author Response
The answer is attached in the file

Reviewer 2 Report
This paper investigated the association between traffic-related noise, ambient air pollution and preeclampsia in Israel. The work can be improved by more rigorous presentations of the results, as well as some explanations and clarifications:
Study design:
-The study recruited 316 pregnant women who underwent prenatal follow up at the clinics in the study area. In total, 285 women agreed to participate in the study. It is interesting to know how these women were representative of the total population in terms of factors such as urban/rural, socioeconomic status, general health conditions. In other words, readers would like to know how results from this study could be extrapolated to other populations.
-It is unclear when the study was conducted. The author mentioned that TRN was assessed between 2013 and 2016; EN was measured between February 2016 and June 2016; it was not mentioned when TRAP was measured. Since TRN was assessed over 3 years, does it mean that for some women TRN (and/or other exposures) was assessed after giving birth or being diagnosed with preeclampsia? More detailed clarification would be helpful.
-The author mentioned that the NOx calculation was based on meteorological data that was measured during 2010-2014. Could the authors explain in more details about how the 2010-2014 data could reflect the pollution status in 2016? Also could the authors help readers understand whether NOx was originally calculated as hourly/daily then aggregated to monthly or the average of pregnancy? Some information in the exposure assessment section was missing and it is hard to follow.
-TRN exposure was self-reported “during pregnancy at enrollment during routine prenatal visits”. Was it overall exposure level during pregnancy or trimester specific? How many times was each woman assessed for TRN?
-EN was assessed at women’s residential address at 4-7pm between February and June 2016. For most study participants how well did the exposure level measured using this method represent the actual exposure level during pregnancy? For example, did most pregnant women stay at home during this time period?
-It is interesting to know how frequently people move in the study location especially for women during pregnancy and whether it is possible to take the change of residential address into account in exposure assessment.
-It was unclear how “combined exposure” was defined in this analysis. Usually an interaction analysis is done to explore the individual and joint effect of two exposures, TRN and TRAP in this case. More extensive analyses are needed before making conclusion on the effect of combined exposures on preeclampsia.
-The reported noise level at residential address may not reflect the actual noise exposure level that women experienced during pregnancy when occupational noise also contributes to the exposure sources. As the authors mentioned in the discussion, the questionnaire was able to capture information on exposure to noise at work, and when this taken into account the correlation between the reported noise level and measured noise level was increased. It would be great to see how results in the main analyses, i.e., logistic regression, are changed when occupational exposures (both TRN and TRAP) are taken into account.
-Since TRN was self-reported, it is important to discuss about potential recall bias including the direction and magnitude of the bias.
Analysis:
-The authors mentioned one-way ANOVA, which is usually used to analyze the difference between the means of more than two groups. In the results, however, it is difficult to know which analysis was performed using this test.
-In multivariate logistic regression, some important socioeconomic status variables such as maternal education, was not adjusted, which makes the effect estimates being sensitive to confounding due to those factors. It would be great if this could be included in the model.
-Table 1:
1). The table shows the distribution of preeclampsia by demographic variables. Could the authors clarify which statistical test was performed?
2). It was mentioned that “Women with preeclampsia had higher number of births (p=0.007).”. However, based Table 1 the mean (SD) number of birth for women with and without preeclampsia was 1.8 (0.8) and 2.8 (1.8), which is reversed from the text. Please clarify.
3). It was confusing how the authors selected certain variables from Table 1 to present in the text. For example, BMI before pregnancy was highlighted (p=0.057) but chronic illness (p<0.001) was not.
4). Current table provides some sense about the association between the outcome of interest and demographic characteristics. It would be great to have a similar table to show the distribution of exposures (TRN/TRAP/EN) by demographic characteristics to help readers understand the association between the exposures and demographic variables.
-In the abstract it was mentioned that the adjusted OR for TRN was 1.2 (p=0.037); OR for NO (adjusted or unadjusted?) was 1.2 (p=0.028). However, in Table 2 and the text, the adjusted OR for TRN was 1.2 (p=0.028); adjusted OR for NO was 1.01 (p=0.031). Numbers in the abstract and results do not match and it is difficult to follow.
-It is unclear why a cut-off point of 6 was selected to categorize a continuous exposure to binary. Could the authors explain possible reasons and how this was comparable with previous studies?
Discussion:
-It is good that the authors discussed about possible reasons for low correlation between the self-reported noise level and the measured noise level. Since the self-reported exposure may only “partially” reflect the actual traffic-related noise exposure level, variable used in the analysis may not be an accurate proxy of the real exposure, therefore results from the analysis needs to be interpreted carefully. Any causal inference should be avoided.
-It was mentioned that “to test the effect of air pollution on the model, any degree increase in the traffic noise scale added a 23% increased risk for preeclampsia.”. It is unclear whether the authors were referring to OR=1.23 (1.1, 1.4) here and what “any degree increase” means.
-The authors mentioned some previous studies in the discussion; however, the current study was not compared with those studies in terms of study design and results. It is hard to know whether the current study is considered as being in line with previous studies from the authors’ perspectives.
Author Response
Answers are in the attached file

Round 2
Reviewer 2 Report
-In the results the authors added “about 5 times higher rate of chronic diseases” for women with preeclampsia. The wording is confusing. Would it be possible to revise into something like “higher percentage of having chronic diseases” or “about 5 times of percentage of having chronic diseases”?
-It is nice that the authors clarified what combined exposures means. However, since the 95%CI is very wide probably due to very few number of women being exposed to both exposures, I hope that the authors could take this into account in the discussion. This type of combined exposure only captures the “strongest” effect of both exposures compared to women who were unexposed to either exposure, therefore it is reasonable to see a “significant” OR. However, like I mentioned in the first round of comments, usually when multiple environmental exposures are investigated, when researchers try to study the “combined” or “joined” effect of two exposures, at least an analysis with interaction is performed to look at 1) only exposed to exposure A; 2) only exposure to exposure B; 3) exposed to both A and B, compared to unexposed to either A or B. Also in this manuscript, it is unknown how many women were classified as being exposed to TRN and/or TRAP using the arbitrary cut-off points. Therefore it is very difficult to interpret the effect estimates.
-Numbers should be rounded to the same number of digits. For example, in the results it was mentioned that “traffic-related noise was found as an independent risk factor for preeclampsia (adjusted OR=1.2, 95% CI 1.1-1.4; p=0.028).”. In Table 2 it was “1.23 (1.1, 1.4)”. It would be easier for readers to follow if numbers are rounded into the same number of digits.
-In Table 2, the authors added “OR=1.01 (0.99, 1.1, P=0.031)” for TRN. The 95%CI included 1 which should yield a non-significant P-value. I’m confused by this inconsistency between 95%CI and P-value.
-I hope that in the abstract 95%CIs could be added in addition to the P-values.
-It was nice to see that the authors clarified details in the cover letter. I was hoping that these could also be reflected in the revised version of the manuscript, such as representativeness of the study population, the assumptions about no exposure change during pregnancy, no change of residential address, mostly staying at home during 4-7pm for study participants, and no change of results by taking occupational exposures into account, etc. It will help readers to better understand the study.
Author Response
Attached in the letter
